# Models of Sports Management in Fitness Centres. Influence of Sex, Age and Sport Frequency. Linear Models vs. Qualitative Comparative Analysis

**Fernando García-Pascual** [1,*], **Carlos Pérez-Campos** [2], **Joaquín García Sánchez** [2], **Ana Soto-Rubio** [1] 
**and Sergio Aguado Berenguer** [1]

1 Physical Education and Sports Department, University of Valencia, 46010 Valencia, Spain; ana.soto@uv.es (A.S.-R.); sergio.aguado@uv.es (S.A.B.)
2 Department of Teaching and Learning of Physical, Plastic and Musical Education, Campus Capacitas, University Catholic of Valencia, 46110 Godella, Spain; Carlos.perez@ucv.es (C.P.-C.); joaquin.garcia@ucv.es (J.G.S.)
* Correspondence: fernando.garcia-pascual@uv.es

**Abstract:** Knowing the perceptions of users of sports services has always been an object of analysis within sports management. This paper attempts to analyse what influences the satisfaction and future intentions of fitness centre customers, beyond management variables, by using two different methodologies. A sample of 389 users of a private sports centre was used. Both linear relationships between variables and the combination of sets were analysed using fuzzy set Qualitative Comparative Analysis fsQCA. It is concluded that management variables (service quality, satisfaction and perceived value) are very important for the prediction of management models, but that, according to the interaction methodology between variables, both frequency and sociodemographic variables play an important role in achieving satisfied and loyal users of the sports service. For the prediction of customers' future intentions, within the analysed sets, it is observed that satisfaction and perceived value are the most predictive variables (raw coverage 0.66). Therefore, and as a consequence, a high number of satisfied and loyal users of the service will allow the economic viability of this service to be achieved.

**Keywords:** sport management; satisfaction; future intentions; fuzzy set qualitative comparative analysis; linear models; age; sex; frequency

## 1. Theoretical Framework

### 1.1. The Management of Sports Services

Understanding users' perceptions of sport services has recently become the main focus of sport management literature [1–3]. Through their experiences with these services, users obtain perceptions that allow them to give values that later become behaviours towards the service. Managers of sports services try to ensure that these behaviours acquired by the user are long-lasting and allow the service to be viable [4]. Various studies in the sport management literature have analysed these user perceptions through different variables such as user perceived quality [5,6], user satisfaction [7] or user future intentions, among many others [8]. One of the sports services that has developed most in recent times for the practice of sport and physical activity are sports centres, which translates to spaces as places where all profiles of society have a place, where they can satisfy their sporting needs and contribute to the improvement of their health. At the same time, it has also been seen how these sports centres, where previously there was a large majority of male population, women are increasingly attending and participating in these services, and giving more positive perceptions of the services acquired than men [9]. The substantial growth experienced by these sports centres means that the versatility of the client profile is

increasing, due to the programmes and the health they offer, which is why increasingly more work in the sports management literature has focused on the segmentation of the users of these sports services [10,11]. Within these sports centre services, there are different models whose fundamental criterion to differentiate them is the price. They are classified from low cost centres to premium centres, passing through medium centres, where they differ from each other in the features and services they offer, all related to the price paid by the user to be part of the fitness centre [1].

Listing the multiple benefits of practicing sports is an easy task, since there are many studies that support this statement [12,13]. The reasons for practicing physical activity are innumerable, including physical, psychological and health aspects, among others [14,15]. The practice of physical exercise is one of the main factors associated with the improvement of physical and psychological well-being [16]. Great importance in this context analyzed is the fitness sector, a very established sector in our society having ample range for improvement and innovation. That is why recently, in 2018, Spain ranked fifth in Europe in revenue, all due to the increase in the realization of physical activity by society, amounting to 5,330,000 members of sports centres, increasing by 26% compared to 2017, and with a penetration ratio of 11.4% for that fitness sector during 2018 [17]. In the European context, it accounted for EUR 27.2 billion revenue, slightly surpassing the United States (EUR 26.6 billion), and could become the largest European fitness market in terms of revenue.

### 1.2. The Management Variables of Fitness Centre

One of the major areas of sports management is sports services, a very broad sector that is increasingly present in our society; therefore, in this context led by sports managers and directors, variables are increasingly analyzed and studied in order to achieve a satisfied and loyal customer to the service. For years, the different management variables that make up the models of sports services have been widely analyzed [18,19] and new variables under study have been added through time. From these sports services, through the programmes and services offered, the aim is to retain the user with the service for as long as possible, with [20] arguing that the cost of attracting new customers is higher than the effort to retain those who already enjoy the service. Kotler (2000) [21] argues that a service will succeed if it provides satisfaction and value to the user, coinciding with a conclusion that value influences satisfaction and, in turn, attitudinal loyalty [22]. Within the sport management literature, both variables, satisfaction and value, have a significant relationship with users' intentions [23,24]. Understanding that by obtaining satisfied users through the quality offered by the sports service, together with the process of creation of value that the user perceives when purchasing the service, will consequently lead to a loyal user, which in turn will raise the percentage of user retention with the service.

### 1.3. The Influence of Sociodemographic Variables on Fitness Centres

For some time now, the practice of physical activity in its hedonistic utility has been largely oriented towards health and well-being, which is why we increasingly find older people in sports services, thus justifying that physical activity allows for a better quality of life [25]. Given the potential increase in the number of these sports centres and their versatility, this favours the practice of sport by a large part of society in terms of age; the older the user of these sports services, the better the service is rated [26] and, therefore, the greater the loyalty to the service [27].

This sporting practice, which is becoming increasingly widespread in our society, also affects sex. The female profile is becoming increasingly more frequent in these sports services, Currently, the majority of society seeks both physical and psychological well-being in sport, with women practising physical activity in order to improve their body image and health [28]. Additionally, in the evaluation of the experience with fitness centres, women rate the sports service more positively [29], and in turn, they have more positive intentions for future behaviour towards the service acquired [30].

Another aspect to be considered in the perceptions of users who attend sports centres is through the experiences measured by the frequency of use of these sports services. More frequent visits to sports services imply a greater degree of contact and experience with the service, which leads to a better evaluation of the service. Therefore, more frequent use of the fitness centre significantly reduces the likelihood that the user will drop out of the service [27], and therefore favours retention and thus user loyalty towards the sports service [31].

Within the sport management literature, the models that analyse these services have almost always analysed the relationships that exist between the variables, but in a separate way, linearly [12,26], where the influence of a single variable on another variable is known. On the other hand, there are also works [3] that have analysed the influence of a set of variables on another variable, fsQCA, thus obtaining more detailed information on the prediction of a variable.

This work analyses the perceptions of the users of a sports centre through the management models of these services, and examines the influence of sex, age and sports frequency, by using two different methodologies. In addition, it intervenes in the loyalty of the user, obtaining in its end the economic viability in a sustainable way.

However, based on previous literature, the following hypotheses are described:

**Hypothesis H1 (H1).** *Management variables are the most influential in predicting future intentions.*

**Hypothesis H2 (H2).** *The fsQCA methodology provides more comprehensive and accurate information than linear models.*

**Hypothesis H3 (H3).** *The age and sex of the customers influences their satisfaction and future intentions.*

## 2. Materials and Methods

### 2.1. Participants

The sample of this research was confirmed by 389 users of a privately managed sports centre, having a mean age of 35.89 years (±15.79) and a total of 218 men (56%) and 171 women (44%). In terms of weekly sports practice, only six users (1.5%) go once a week to the sports facility, fifty-five users (13.8%) are those who practice sports activity in the facility twice a week, one hundred and sixty-three (41%) of the surveyed users go three times a week to the sports complex; finally, one hundred and seventy-four users (43.7%) stated that they go four times a week or more to the sports centre to practice physical activity. The inclusion criteria for choosing the sample were that the customers who voluntarily participated were of legal age, had to have paid the corresponding fee and frequently attended the sports centre. On the other hand, there was also a reason for exclusion, which was that users could not participate in the study if, due to their language, they had any problems in understanding the questionnaire. The population size was 1950 users, although this sample includes 102 university students who are not fluent in Spanish, giving a reference population of 1848 customers. With a total of 389 users in the sample, the margin of error is 4.42%, with a confidence level of 95% and with sample heterogeneity of 50 %.

### 2.2. Instrument

The measurement instrument, whose scales were validated, consisted of 49 Likert-type indicators with a minimum value of 1 (Strongly Disagree) and a maximum value of 5 (Strongly Agree). The instrument used the scale for measuring perceived service quality (SSQRS) [32], which measured service quality through four dimensions (36 items) such as quality of the programme, quality of the interaction, quality of the outcome and quality of the environment. General satisfaction, composed of two specific items [33], and the perceived value scale composed of three factors (emotional value, price value and

social value) [34]. Future intentions consist of four specific items [35]. Finally, sociodemographic variables such as age and sex were measured, together with the frequency of sport performed in the fitness centre.

### 2.3. Procedure and Data Analysis

The measurement instrument was passed at the entrance of the sports facility itself, organised at different times to be able to collect the perceptions of users with different profiles and also the different sociodemographic variables. As in other studies in the literature with similar characteristics [3], a nonprobabilistic convenience sample was used. Users entering or leaving the facility itself were asked to participate in the survey in order to ask them about their perception of the facility, as the centre had a large number of registered users. The users showed voluntariness during the process of participation in the execution of the questionnaires, and they were also guaranteed the confidentiality of the data. This study was conducted in early 2018.

After determining the perceptions of the users of the sports centre, we proceeded to analyse the different results obtained, for which we first obtained the descriptive data using the SPSS (Statistical Package for the Social Sciences) v.25 programme, in order to determine in more detail the sociodemographic variables of the users of the sports facility. As for the frequency of sports practice, a direct recoding of the set of results was carried out, changing from 1 to 0, from 2 to 0.33, from 3 to 0.66 and from 4 to 1. SPSS was then also used to perform hierarchical regression (HRM) of the different management variables, and to see if the model improves significantly with the inclusion of the sociodemographic variables and sport frequency. Subsequently, the variables were analysed using fsQCA (fuzzy set qualitative comparative analysis), an analysis methodology based on set theory. This technique makes it possible to determine which causal relationships are present in order to arrive at the result. For this purpose, the calibration values of the results obtained were first evaluated, and then, using fsQCA, the set conditions were analysed, the last step being the necessary and sufficient analysis of the different models.

As seen in Table 1, the different scales provided good consistency values, as all gave values above the recommended 0.70 [36] for Cronbach's alpha ($\alpha$).

**Table 1.** Reliability of the scales.

|  | $\alpha$ | N |
| --- | --- | --- |
| Service quality | 0.87 | 36 |
| Overall satisfaction | 0.82 | 2 |
| Perceived value | 0.83 | 7 |
| Future intentions | 0.86 | 4 |

Note: $\alpha$, Cronbach alpha.

## 3. Results

### 3.1. Hierarchical Regression Model

As shown in Table 2, it can be seen that for each model the management variables explain most of the variance with respect to the sociodemographic variables and the sport frequency of the users. In the prediction of user satisfaction, it is observed that in step 1 the management variables (SQ and PV) explain 35% ($\Delta R = 0.35$; $p < 0.001$) of the model, while they do not show significance ($\Delta R = 0.01$; $p > 0.05$) in either step 2 (inclusion of age and sex) or in step 3 (frequency). In contrast, as for the future intentions model, the three steps analysed account for 47% ($p < 0.001$). As in the previous models, the management variables, service quality (SQ), satisfaction (SAT) and perceived value (PV), in the first step explain 46% ($\Delta R = 0.46$; $p < 0.001$) of the variance, while, in the following steps with the inclusion of sex, age and sport frequency, the variance of the model increases by 1% ($\Delta R = 0.01$; $p > 0.05$). Within the last step analysed, it is observed that of the management variables, perceived value ($\beta = 0.40$; $p < 0.001$) offers the greatest predictive weight on future

intentions, followed by satisfaction (β = 0.24; *p* < 0.001) and perceived quality (β = 0.12; *p* < 0.05). Table 2 shows the 3-step analysis with hierarchical regression.

**Table 2.** Hierarchical regression models of management variables, age, sex and sport frequency.

|  | Satisfaction | | Future Intentions | |
|---|---|---|---|---|
|  | $\Delta R^2$ | $\beta$ | $\Delta R^2$ | $\beta$ |
| **Step 1** | 0.35 *** |  | 0.46 *** |  |
| Service quality |  | 0.38 *** |  | 0.13 * |
| Satisfaction |  |  |  | 0.25 *** |
| Perceived value |  | 0.26 *** |  | 0.41 *** |
| **Step 2** | 0.01 |  | 0.01 ** |  |
| Service quality |  | 0.38 *** |  | 13 * |
| Satisfaction |  | - |  | 0.25 *** |
| Perceived value |  | 0.26 *** |  | 0.40 *** |
| Frequency |  | −0.04 |  | 0.03 |
| **Step 3** | 0.01 |  | 0.01 |  |
| Service quality |  | 0.37 *** |  | 0.12 * |
| Satisfaction |  | - |  | 0.24 *** |
| Perceived value |  | 0.27 *** |  | 0.40 *** |
| Frequency |  | −0.30 |  | 0.04 |
| Age |  | −0.01 |  | 0.05 |
| Sex |  | −0.05 |  | −0.07 |
| Total $R^2_{adj}$ | 0.35 *** |  | 0.47 *** |  |

Note: $\Delta R^2$, R-square change; β, standardized beta; $R^2$ adj, R-square adjusted; * $p \leq 0.05$; ** $p \leq 0.01$; *** $p \leq 0.001$.

### 3.2. Qualitative Comparative Analysis (fsQCA)

First, before performing the two analyses, as shown in the table above, the descriptive statistics and calibration values with the three defined thresholds were obtained (Table 3).

**Table 3.** Descriptive statistics and calibration values.

|  |  | SQ | SAT | PV | FI | AGE |
|---|---|---|---|---|---|---|
| *n* |  | 389 | 389 | 389 | 389 | 389 |
|  |  | 0 | 0 | 0 | 0 | 0 |
| Mean |  | 361,940.89 | 17.54 | 18,774.54 | 327.81 | 35.89 |
| SD |  | 362,997.07 | 5.70 | 20,976.67 | 197.82 | 14.53 |
| Minimum |  | 1.50 | 1.00 | 1.00 | 8.00 | 18.00 |
| Maximum |  | 1,562,500 | 25 | 78,125 | 625 | 77 |
|  | 10 | 46,210.20 | 9.00 | 1728.00 | 81.00 | 20.00 |
| Percentiles | 50 | 245,367.50 | 16.00 | 11,520.00 | 256.00 | 32.00 |
|  | 90 | 886,936.25 | 25.00 | 50,000.00 | 625.00 | 58.00 |

Note: SD, standard deviation; SQ, service quality; SAT, satisfaction; PV, perceived value; FI, future intentions.

Next, a necessity analysis was conducted to determine whether any of the conditions are necessary for both high and low levels of user satisfaction and future intentions. As can be seen in Table 4, none of the conditions are necessary, as the consistency of each of them does not show values above 0.90 [37].

Finally, the sufficiency analysis was carried out in order to determine whether the conditions of the model are sufficient. As can be seen in Table 5, which shows the most important combinations of conditions of the intermediate solution, both the high and low levels of satisfaction (0.85; 0.81) and future intentions (0.84; 0.83) of the users are adequate since they have consistency values greater than 0.75 [38].

**Table 4.** Necessity analysis for satisfaction and future intentions.

|  | SAT | | ~SAT | | FI | | ~FI | |
|---|---|---|---|---|---|---|---|---|
|  | **Cons** | **Cov** | **Cons** | **Cov** | **Cons** | **Cov** | **Cons** | **Cov** |
| SQ | 0.71 | 0.88 | 0.43 | 0.41 | 0.72 | 0.82 | 0.43 | 0.45 |
| ~SQ | 0.52 | 0.54 | 0.87 | 0.70 | 0.52 | 0.50 | 0.83 | 0.73 |
| SAT | - | - | - | - | 0.82 | 0.76 | 0.55 | 0.47 |
| ~SAT | - | - | - | - | 0.43 | 0.51 | 0.72 | 0.78 |
| VP | 0.68 | 0.86 | 0.44 | 0.44 | 0.73 | 0.86 | 0.39 | 0.42 |
| ~VP | 0.55 | 0.56 | 0.86 | 0.68 | 0.51 | 0.48 | 0.87 | 0.75 |
| FREQ | 0.84 | 0.63 | 0.86 | 0.50 | 0.86 | 0.60 | 0.84 | 0.53 |
| ~FREQ | 0.33 | 0.75 | 0.36 | 0.65 | 0.32 | 0.68 | 0.36 | 0.70 |
| AGE | 0.60 | 0.70 | 0.58 | 0.53 | 0.62 | 0.68 | 0.55 | 0.54 |
| ~AGE | 0.60 | 0.64 | 0.67 | 0.56 | 0.58 | 0.59 | 0.68 | 0.62 |
| SEX | 0.52 | 0.54 | 0.58 | 0.46 | 0.52 | 0.50 | 0.57 | 0.50 |
| ~SEX | 0.48 | 0.59 | 0.42 | 0.41 | 0.48 | 0.56 | 0.43 | 0.44 |

Note: ~, absence of condition; Con, consistency; Cov, coverage; SQ, service quality; SAT, satisfaction; PV, perceived value; FI, future intentions; FREQ, sport frequency; SEX, Condition needed: consistency ≥ 0.90; –, do not calculate according to the theoretical model.

**Table 5.** Main conditions of sufficiency analysis for satisfaction and future intentions (intermediate solution).

|  | SAT | | | ~SAT | | | FI | | | | ~FI | | | |
|---|---|---|---|---|---|---|---|---|---|---|---|---|---|---|
|  | Consistency cutoff 0.87 | | | Consistency cutoff 0.83 | | | Consistency cutoff 0.87 | | | | Consistency cutoff 0.85 | | | |
|  | 1 | 2 | 3 | 1 | 2 | 3 | 1 | 2 | 3 | 4 | 1 | 2 | 3 | 4 |
| Service quality | ● |  |  | O | O | O |  | ● |  | ● | O |  | O | O |
| Satisfaction |  |  |  |  |  | ● |  |  |  | ● | O | O |  | O |
| Perceived value |  | ● | ● |  | O | O | ● | ● | ● |  | O | O | O |  |
| Frequency |  |  |  | O | O |  | ● |  |  |  | O |  |  |  |
| Age |  |  | ● |  | ● | ● |  |  | ● | ● |  | O | O | O |
| Sex |  | O |  | ● |  | ● |  |  |  |  |  |  | ● | ● |
| Raw coverage | 0.71 | 0.32 | 0.46 | 0.17 | 0.23 | 0.26 | 0.66 | 0.55 | 0.50 | 0.47 | 0.63 | 0.20 | 0.33 | 0.29 |
| Unique coverage | 0.23 | 0.02 | 0.02 | 0.06 | 0.12 | 0.14 | 0.03 | 0.01 | 0.01 | 0.01 | 0.26 | 0.01 | 0.05 | 0.02 |
| Consistency | 0.88 | 0.88 | 0.90 | 0.86 | 0.86 | 0.82 | 0.90 | 0.92 | 0.91 | 0.91 | 0.87 | 0.90 | 0.86 | 0.86 |
| Solution consistency |  | 0.85 |  |  | 0.81 |  |  | 0.84 |  |  |  | 0.83 |  |  |
| Solution coverage |  | 0.78 |  |  | 0.43 |  |  | 0.80 |  |  |  | 0.70 |  |  |

Note: ~, absence (low levels) of condition; ● presence (high levels) of condition; O bsence (low levels) of condition. SAT, satisfaction; FI, future intentions; –, do not calculate according to the theoretical model. Expected vector for satisfaction: 1.1.1.1.1 (0: absent; 1: present), expected vector for ~ satisfaction: 0.0.0.0.0 (0: absent; 1: present), expected vector for future intentions: 1.1.1.1.1.1. (0: absent; 1: present), expected vector for ~ future intentions: 0.0.0.0.0.0 (0: absent; 1: present).

First, three combinations of conditions for the presence of SAT are presented, these are SQ (raw coverage: 0.71; consistency: 0.88), PV*~SEX (raw coverage: 0.32; consistency: 0.88) and PV*AGE (raw coverage: 0.46; consistency: 0.90), all of which explain 78% of high levels of user satisfaction (solution coverage: 0.78; solution consistency: 0.85). On the other hand, three combinations of conditions ~SQ*~FREQ*SEX (raw coverage: 0.17; consistency: 0.86),

~SQ*~PV*~FREQ*AGE (raw coverage: 0.23; consistency: 0.86) and ~SQ*~PV*AGE*SEX (raw coverage: 0.26; consistency: 0.82) explain 43% of low levels of satisfaction (solution coverage: 0.43; solution consistency: 0.81).

Finally, four combinations of conditions explain 80% of the FI (solution coverage: 0.80; solution consistency: 0.84); these combinations are SAT*PV (raw coverage: 0.66; consistency: 0.90), SQ*PC*FREQ (raw coverage: 0.55; consistency: 0.92), PV*AGE (raw coverage: 0.50; consistency: 0.91) and SQ*SAT*AGE (raw coverage: 0.47; consistency: 0.91). Meanwhile, four combinations are also given for the absence of FI: ~SQ*~SAT*~PV (raw coverage: 0.63; consistency: 0.87), ~SAT*~PV*~FREQ*~AGE (raw coverage: 0.20; consistency: 0.90), ~SQ*~PV*~AGE*SEX (raw coverage: 0.33; consistency: 0.86) and ~SQ*~SAT*~AGE*SEX (raw coverage: 0.29; consistency: 0.86), thus explaining 70% of the low FI levels (solution coverage: 0.70; solution consistency: 0.83).

## 4. Discussion

Sports centres, as a sports service, have become well established in recent management literature [3,18]. This work tries to discover what future behaviour users will have after using the services of a fitness centre. This was done by analysing their perceptions using two different methodologies. Physical activity and sport as a prelude to health and well-being have allowed many different constituents of our society to attend sports centres. Given the change in the trend of these sports services, this has occurred in recent years without any distinction and in a heterogeneous manner.

This is demonstrated by the fact that, increasingly, and given the amount of demand for these sports services, there are works within the sport management literature, where the users are analysed in a segmented way, in order to know in greater depth their perceptions of the service acquired [10,11].

It is also found in the literature how the different management models that have prevailed in these fitness centres have been analysed, and how knowing the perceptions of the users helps the viability of the service itself [39]. Knowing the future behaviour of the users of the sports centre has always been a key factor in sports management, and loyalty is fundamental for the viability of these sports services. Variables such as service quality, perceived value and consumer satisfaction have been widely analysed in these contexts [1,23,31], and their influence on the future intentions of users is thus directly known. Another variable that can indicate whether higher levels of loyalty can be obtained is to know the influence of the frequency with which users attend the sports service centre on user loyalty, as there is a direct positive relationship between the user's sports frequency and the user's intentions to return to the service [27].

These studies found in the literature are usually related through linear models, i.e., the direct influence of one variable on another, but do not attempt to analyse the combinations and interactions of the variables [38]. In this work, the results are also analysed through fsQCA methodology, by means of which a result is obtained through combinations of attributes [37]. Within the sport management literature, increasingly more works analyse sport services through this methodology [40,41]. Consequently, this paper attempts to analyse how the satisfaction and future intentions of the users of a fitness centre are predicted through a linear methodology and a combination of conditions [40,41].

Concerning the linear methodology, i.e., hierarchical regression, it is observed that the management variables are those that significantly obtain the majority of the predictive weight, both for user satisfaction and future intentions, results that have also been observed in the literature [3,41]. Hypothesis 1 can be supported. This may be so, because in linear regression there is only one explanatory process, only one path, and the management variables are the ones, although not the only ones, that offer the best prediction. In the prediction of customer satisfaction with this methodology, it can be seen that the quality of service is the one that offers the greatest predictive weight. However, in the prediction of future intentions, the perceived value of customers is the variable that most significantly

influences this loyalty. Likewise, our results show that neither age, gender nor sport frequencies have a significant influence on customer satisfaction or future behaviour.

The fsQCA methodology, in contrast, defines that in both models there is no condition necessary for prediction, as it does not obtain the values found in literature [37]. Observing each model of this methodology and in contrast to the regression, it can be seen that sport frequency, together with age and sex, play a more influential role in each model. Therefore, Hypothesis 2 can be supported. It can be seen how, for example, to obtain both high and low levels of satisfaction, the variables of frequency, age and sex do appear and have an influence on most of the paths that make up the prediction of this variable, which is especially important for sport management. The same situation is found in the combination of sets for predicting future user behaviour. For example, it is observed how fitness centre managers can obtain high levels of satisfaction and high perceived value of service from participating women and all those of advanced age. These relationships are also found in the literature [18,26]. Hypothesis 3 can be supported. In addition, male users with low satisfaction and young people tend not to repeat the service purchased. Within the literature, there are studies that also argue that it is men who give lower rating levels than women [42], although it is true that there are studies [43] that do not show significance in the sociodemographic variables. However, for high levels of intention to repurchase the service, this would be reflected in those users with high levels of perceived value and with advanced age. Perhaps the fact that young people are less likely to repeat service is due to their economic situation, as the labour market is a major concern for them, in contrast older people who have a stable job or a pension and are more likely to stay longer in the service. It is also observed that a low sport frequency in the fitness centre is a characteristic of low levels of satisfaction and intention to repeat the service. This fact is also found in the literature, [31] where it is argued that sport frequency affects the retention of sport service customers.

Therefore, using both methodologies in a complementary way [40] helps sport managers to have more accurate information concerning users' perceptions of the service. Since the purpose of these services and the sports managers is customer retention and loyalty, accurate information is essential for those who manage fitness centres, particularly in a context where the offer is very high and changing.

## 5. Conclusions

Ability to know the perceptions of the users of sports services has recently become one of the objectives of researchers. Through the opinions of users of a sports centre, this paper attempts to analyse the influence of sociodemographic variables and sport frequency on users' future behavioural intentions by using two different methodologies.

First, it is observed that the use of two different methodologies provides more accurate and consequently more valuable results to understand the perceptions of the customers of these sports entities. Likewise, it can be seen how, in a linear model, the management variables (quality of service, satisfaction and perceived value) are those that offer the greatest significance in the relationships.

Second, application of fsQCA methodology shows that other variables such as sociodemographic variables or sport frequency also influence the prediction of the variables. It is observed that factors such as older customers or a high frequency of attendance to the sports service also reinforce the positive future behaviour of users towards the sports service.

Third, it is also confirmed that male or younger gym attendees are less likely to return to the sports service in the future. This indicates that targeting and analysing groups with low levels of loyalty (young clients and men) in order to adapt services, improve perceptions, and promote higher sport frequency, will help to improve future client intentions in an overall positive way.

## 6. Implications, Limitations and Future Lines of Research

After analysing and discussing the results and conclusions, different management implications are provided. First, this work contributes to the literature on sports management, incorporating information that allows the main objective of these sports entities, the retention of customers, to be achieved. Two different but complementary methodologies were used to analyse these perceptions, which allow the managers of these sports centres to gain a more detailed understanding of user behaviour. On a practical level, this allows these fitness centre managers to work on those aspects (quality, programmes, environment, satisfaction, etc.) where perceptions have reached low levels, and to be able to act in a very specific way in order to increase these levels and retain constituents of all profiles that come to the sports service for as long as possible.

This work, by obtaining results that analyse age and sex, also involves managers in establishing new strategies and actions, which will allow constituents of all characteristic profiles to have high levels of satisfaction and future intentions to return.

In terms of limitations and future lines of research, it is found that a nonprobabilistic sample has been used, so the population is not universally represented, although on observing the sample analysed, it is estimated that there is an approximation to the object of study. It is also necessary to analyse different sports centres with similar characteristics, in order to determine if the results are homogeneous. This study focuses mainly on one sports centre, making it difficult to generalise it to other sports centres with other characteristics and singularities. In future lines of research, studies could also focus on public sports centres, where differences with private sports centres are likely to be found. It would also be proposed to increase the sample size and carry out a probability sample.

**Author Contributions:** Introduction, F.G.-P.; software, C.P.-C.; method, F.G.-P., S.A.B. and A.S.-R.; procedure, J.G.S. and C.P.-C.; results, F.G.-P., A.S.-R. and J.G.S.; discussion, S.A.B. and C.P.-C. All authors have read and agreed to the published version of the manuscript.

**Funding:** This research received no external funding.

**Institutional Review Board Statement:** Not applicable.

**Informed Consent Statement:** Informed consent was obtained from all subjects involved in the study.

**Data Availability Statement:** Not applicable.

**Conflicts of Interest:** The authors declare no conflict of interest.

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
