# Peer review of "Models of Sports Management in Fitness Centres. Influence of Sex, Age and Sport Frequency. Linear Models vs. Qualitative Comparative Analysis"

_sustainability, doi:10.3390/su13168995_

Round 1
Reviewer 1 Report
My recommendations are the following:
In the title to mention what HRM vs. QCA.
I recommend you to describe descriptively what it represents - fsQCA; from keywords.
In the abstract, the method applied is not clear, to whom, for whom, this study is addressed, I recommend clarification. Mention what variables the study refers to.
In section 3.1. I recommend you to maintain what were the inclusion and exclusion criteria of the subjects.
In section 3.2 I recommend you to mention which questionnaires you applied, also to calculate the reliability of the questionnaires or groups of items.
I recommend mentioning when the study was conducted.
Section 4.1. forward on the first table, What are the management variables (SQ, SAT, PV)? You mention them under table 3, but these abbreviations appear before. I recommend the correction.
I recommend that you clearly specify which management models to consider in your study. And to mention what they consist of as a form of organization by sex and age. According to the information, you are referring to a single private center, it is not clear what you are looking for in terms of SQ, etc.
You are missing the conclusions section.
Also the discussion section should be based on the results of the present study taking into account previous studies. You also do a theoretical substantiation similar to the one from the introduction. I recommend rewriting.
Following the revision of this article, I recommend that the abstract be rewritten more clearly and focused on the topic.
Author Response
Dear Reviewer,
Thank you for your suggestions on the manuscript, they have
improved the content of the paper, and as authors they have also been
of great help to us. After reviewing your suggestions, the title has
been modified as well as the keywords, describing the methodologies
used. The abstract has been modified. Within the methodology, the
reliability of the instrument used has been analysed. Also, a section
on conclusions has been added as suggested. The suggestions made
by you have been rewritten and modified.

Reviewer 2 Report
The authors make an interesting and practical article for sports services. However, certain changes are required to make the content of the paper more understandable.
The abstract does not include the relevant data of the study, it only generalizes the results. Eliminate unnecessary information from it (because it is in the text) and include more concrete and relevant results. Be careful with the language and structure of the English sentences, do not use sentences that are so long (as a consequence of having translated the paper from Spanish to English) and grammatically incorrect.
I don't understand why it makes an introduction and a theorical framework. Some papers do it but it is usually not the most appropriate way. Please unify these two sections. Organize the information appropriately:
- A first part that addresses the general importance of the topic (it has already been done),
- Analyze the state of the current scientific literature in relation to the approach addressed (it is the most important but weakest part of its introduction),
- Explain what is the lack of current scientific literature and why it is necessary to study it,
- Finalize with the objectives of the study and include the hypotheses.
From the method to the results, the decimals are not unified. Use only one decimal place across the entire paper. I remind you that in English "." not ",". The p-values are sometimes put with the "0" before the point and sometimes not, why?
From the discussion onwards the information is confusing. You must structure the paper from here on. At the beginning of the discussion, remember the objective of the study and then compare the results of your variables with the results of other studies. First recall your outstanding results, contrast them with the existing literature (this part needs to be expanded) and then make your interpretation of the differences, if any. Currently you mix results of your study, the results of other studies, limitations and conclusions. You must order all the information.
When you finish the discussion, include a different section with the most relevant conclusions of your study. Write them clearly.
Include subsections in the conclusions: implications and limitations / future lines of research.
English language should be carefully reviewed throughout the entire paper. The authors use excessively long sentences with many subordinate sentences that make it difficult to understand the text.
Author Response
Dear Reviewer,
Thank you for your suggestions on the manuscript, they have
improved the content of the paper, and as authors they have also been
of great help to us. After reviewing your suggestions, the abstract and
the introduction have been modified. We have also repositioned the
objectives and added the hypotheses of our work. The discussion has
been restructured and the conclusions section has been added, as well
as implications, limitations and future lines of research.

Reviewer 3 Report
Dear authors, I attach my comments
Title
In the title, change gender (cultural) to sex (genetic).
HRM vs QCA: do not use acronyms that are not internationally known.
Abstract
The abstract includes a brief contextualisation, the objective and the main findings. However, neither the sample selected nor the methodology used in the study is clearly specified.
Keywords:
It is recommended to change gender (cultural) to sex (genetic).
Introduction
Change the word gender to sex.
Theoretical Backgrounds
Change the word gender for sex.
I believe that the studies provided are insufficient, as well as the lack of interesting concepts such as: types of fitness centres, types of users, types of activities...
In addition, it is necessary to make a clear exposition of the situation of fitness centres between Europe and USA and with Spain in this specific case.
HRM vs QCA: these concepts should be explained in the theoretical framework.
Materials and Methods
Participants
- The sample is neither representative nor extrapolable. This research is a case study.
- The justification why? There are 389 respondents, are they the total?
- There are basic inaccuracies such as: size of the universe, % heterogeneity of the sample, margin of error, confidence level... that do not appear.
Instrument
It is not indicated whether a developed and validated instrument is used or not.
Justify this with references to other studies that have used the same: A non-probabilistic convenience sample was used.
Results
Differentiation by sex is necessary for comparison. Also by age range would be ideal.
Table 4: cut and with errors at the bottom of the table.
Discussion
Everything indicated in the theoretical framework section should be included.
Conclusions
Not stated. Should be included in a clear manner.
Managerial Implications
In management articles, 4-5 proposals derived from research must be put forward in order to be applied in real life.
Limitations and Future Lines of Research
Here you should indicate the limitations in carrying out the research and what could not be solved.
I recommend incorporating a future line of research that gives continuity to this article.
Author Response
Dear Reviewer,
Thank you for your suggestions on the manuscript, they have
improved the content of the paper, and as authors they have also been
of great help to us. After reviewing your suggestions, the title has
been modified, both the abbreviations of the methodologies and
changing gender for sex. The abstract has been modified, naming also
the methodology used. In the method section, the limitations have
been specified, as well as the population size and margin of error. The
validations of the scales used in the instrument have also been
indicated. A section on conclusions has been added, as well as a
section on management implications, and the limitations found in the
work, as well as possible future lines of research.
Round 2
Reviewer 1 Report
no comments
Author Response
Thank you for the above corrections, which help us to improve our research work.
Reviewer 2 Report
The authors make an interesting and practical article for sports services. However, certain changes are required to make the content of the paper more understandable.
The abstract continues to be unsuitable, include more concrete and relevant results and be careful with the language and structure of the English sentences, do not use sentences that are so long (as a consequence of having translated the paper from Spanish to English) and grammatically incorrect.
The fisrt part is introduction no Theorical framework.
The parts of the introduction must be structurally divided and the information described does not go from the most general to the most specific, you only write disorderly information. In the case that you consider that it is ordered, explain it in the letter to the reviewer. Check out this comment from the previous review.
Why do they continue with a different number pattern? Decimal with point, number of decimals, p-values are sometimes put with the "0" before the point and sometimes not, why?
You have improved the discussion somewhat but AT THE BEGINNING of the discussion, remember the objective of the study and then compare the results of your variables with the results of other studies. First recall your outstanding results, contrast them with the existing literature (this part needs to be expanded) and then make your interpretation of the differences, if any. Always use this structure for each of the variables discussed in the section. Structure well.
Remember.... "When you finish the discussion, include a different section with the most relevant conclusions of your study. Write them clearly". You cannot start clear conclusions with a subjective comment: "After observing the results, different conclusions have been drawn ..." The conclusions of a scientific work must be objectively written and only extracting the conclusions of the study.
Also remember. English language should be carefully reviewed throughout the entire paper. The authors use excessively long sentences with many subordinate sentences that make it difficult to understand the text.
Author Response
Dear reviewer,
Again thank you for your considerations of our work which helps us to improve our
research work. Therefore, after reviewing your considerations, we would like to explain
the modifications made:
- The abstract has been modified, but we cannot put more results, as the word limit does
not allow us to do so. Some results are included.
-We put theoretical framework, because another reviewer told us not to put the
introductory title, but to put one or the other.
-We have structured the theoretical framework, and we understand that this section goes
from the general to the specific. First talking about sport services in general, then sport
centres, and then the variables that are analysed within the literature. But if you think it
is insufficient or you think we should modify it, tell us and we will do it.
-We have unified the numerical pattern
-We have added interpretations of the results in the discussion, which we have observed
that what you were telling us we have done, put the objective of the work first, also put
the main results and compare them with the literature.
We have re-structured the conclusions

Reviewer 3 Report
Dear authors, I attach the comments I made and which have not been considered
Abstract - These changes have not been considered:
The abstract includes a brief contextualisation, the objective and the main findings. However, the sample selected used in the study is not clearly specified.
Theoretical Backgrounds - These changes have not been considered:
I believe that the studies provided are insufficient, as well as the lack of interesting concepts such as: types of fitness centres, types of users, types of activities...
In addition, it is necessary to make a clear exposition of the situation of fitness centres between Europe and USA and with Spain in this specific case.
HRM vs QCA: these concepts should be explained in the theoretical framework.
Materials and Methods - These changes have not been considered:
Participants
- There are basic inaccuracies such as: % heterogeneity of the sample, confidence level... that do not appear.
Author Response
Dear reviewer,
Again thank you for your considerations of our work which helps us to improve our research work. Therefore, after reviewing your considerations, we would like to explain the modifications made:
- The sample used in our work has been specified in the abstract.
-In the theoretical framework, a brief explanation of the types of fitness centres that exist has been added, as well as a comparison with the European and US markets. At the end of the theoretical framework, there is already a brief explanation of the two methodologies, as well as within the data analysis section there is also an explanation, but if you want us to expand it further, let us know and we will do so.
-The confidence level and the % of heterogeneity of the sample have been specified in the manuscript.

Round 3
Reviewer 2 Report
Dear authors, thank you for making part of the changes. Carrying out these together with the rest of the reviews has substantially modified your article. However, they must make the following modifications:
- Dear authors, regarding the abstract, you should include the most relevant numerical results of your work and eliminate theoretical information (from the beginning) if you need it to stick to the number of words.
- The discussion should begin with the reminder of the objective, not with subjective information that is not quoted: "A few years ago, these sports facilities were seen as places ..." This is what I was referring to in the previous comment.
- In an article with analysis features like yours, I still don't see the reminder of the results in the discussion. Please refer to the statistical level.
Best regards and once again congratulations.
Author Response
Dear reviewer,
Once again, thank you for your considerations that help us to improve our article, and consequently to improve our research work. Thank you.
Therefore, after reviewing your considerations, we would like to explain the
modifications made:
- We have modified the entire abstract, eliminating theoretical information and adding some numerical values from the sufficiency analysis, from the methodology that has provided us with more information, such as the fsQCA.
- We have modified and restructured the beginning of the discussion, as you have indicated.
-We have added statistical information within the discussion based on our results. We have added mainly the results of the linear model, which was not reflected. As for the fsQCA results in the discussion, we have expanded them a little more, although it is true that the argumentation of these results was more extended in the discussion.
We hope that we have been able to correct your considerations correctly.

Reviewer 3 Report
Dear authors, I indicate my comments:
Theoretical background
Lines 45-49: Relevant references should be added.
Materials and Methods
Instrument
The validation process should be indicated. Example:
Sánchez-Sáez, J.A.; Segado Segado, F.; Calabuig-Moreno, F.; Gallardo Guerrero, A.M. Measuring residents' perception of corporate social responsibility in small and medium-sized sporting events. Int. J. Environ. Res. Public Health 2020, 17, 8798. https://doi.org/10.3390/ijerph17238798
Justify this with references to other studies that have used the same. Line 151: "A non-probability convenience sample was used".
Conclusions
Line 327: error, "intervene".
Author Response
Dear reviewer,
-Our instruments used had already been validated in the original paper. Therefore, we did not include a validation process for the instrument. I don't know if you are referring to this validation.
-We have justified with references to another study, the reason for the non-probabilistic convenience sample.
-We have modified within the conclusions the “intervene” by “influence”.
We hope that we have been able to correct your considerations correctly.
